# Shear Wave Elastography in Bruxism—Not Yet Ready for Clinical Routine

**DOI:** 10.3390/diagnostics13020276

**Published:** 2023-01-11

**Authors:** Cem Toker, Justus Marquetand, Judit Symmank, Ebru Wahl, Fabian Huettig, Alexander Grimm, Benedict Kleiser, Collin Jacobs, Christoph-Ludwig Hennig

**Affiliations:** 1Department of Epileptology, Hertie-Institute for Clinical Brain Research, University of Tübingen, 72076 Tübingen, Germany; 2Department of Neural Dynamics and Magnetoencephalography, Hertie-Institute for Clinical Brain Research, University of Tübingen, 72076 Tübingen, Germany; 3MEG-Center, University of Tübingen, 72074 Tübingen, Germany; 4Department of Orthodontics, University Hospital Jena, 07743 Jena, Germany; 5Department of Prosthodontics, University Clinic for Dentistry, Oral Medicine, and Maxillofacial Surgery, University of Tübingen, 72076 Tübingen, Germany

**Keywords:** bruxism, SWE, musculus masseter, temporomandibular dysfunction, elastography in dentistry

## Abstract

Ultrasound shear wave elastography (SWE) is an emerging modality for the estimation of stiffness, but it has not been studied in relation to common disorders with altered stiffness, such as bruxism, which affects almost one-third of adults. Because this condition could lead to an increased stiffness of masticatory muscles, we investigated SWE in bruxism according to a proof-of-principle and feasibility study with 10 patients with known bruxism and an age- and gender-matched control group. SWE of the left and right masseter muscles was estimated under three conditions: relaxed jaw, 50% of the subjective maximal bite force, and maximal jaw opening. Rejecting the null hypothesis, SWE was significantly increased during relaxed jaw (bruxism 1.92 m/s ± 0.44; controls 1.66 m/s ± 0.24), whereas for maximal mouth opening, the result was vice versa increased with 2.89 m/s ± 0.93 for bruxism patients compared with 3.53 m/s ± 0.95 in the healthy control, which could be due to limited jaw movement in chronic bruxism patients (bruxism 4.46 m/s ± 1.17; controls 5.23 m/s ± 0.43). We show that SWE in bruxism is feasible and could be of potential use for diagnostics and monitoring, though we also highlight important limitations and necessary methodological considerations for future studies.

## 1. Introduction

Ultrasound shear wave elastography (SWE) is an emerging modality for the estimation of muscle elasticity or stiffness due to various conditions, such as neuromuscular diseases [1,2]. Apart from neuromuscular diseases, other disorders lead to an alteration of muscle stiffness but are not in the scope of clinicians who perform ultrasounds on a daily basis. Bruxism refers to teeth grinding or teeth clenching, and presents as a craniomandibular dysfunction. A total of 20% of adults have bruxism symptoms, which are expressed differently in the symptomatic [3]. The rhythmic masticatory muscle activity is particularly high in bruxism at night, for this reason it is often referred to as sleep bruxism, which is said to be caused by the transient activity of the brainstem arousal-reticular ascending system [4,5]. It is a disease of the oral, maxillofacial area, which has various effects and symptoms: besides pain during chewing, tooth attrition, and muscle changes such as hypertrophy, the main symptom is the stiffness of the masticatory muscles resulting in a limited mouth opening [6,7,8]. This stiffness of the masticatory musculature and its pathological changes cannot be measured in a uniform diagnostic manner [9]. One of these disorders is bruxism, which is a common condition that affects almost one-third of adults [3,10]. The diagnostics and monitoring of bruxism rely on the history of pain and tension of the masticatory muscles, and an error-prone clinical examination of the masticatory muscles, temporo-mandibular joints, and dental status according to international harmonized diagnostic criteria for temporo mandibular disorders (DC/TMD) [11]. Herein, the active and passive mouth opening measured as the distance between the incisal edges of the central incisors acts as a surrogate for stiffness of the muscles.

One way to objectively measure changes in masticatory muscles, preferably the M. masseter, could be ultrasound SWE. Ultrasound can easily visualize the size change of the masseter muscles for diagnosis and therapy [12]. SWE is a recently developed method of ultrasound diagnostics [13], and the basic principle is generally as follows: shear waves (or transversal waves) are acoustically induced by a transducer and propagate perpendicular to the main longitudinal transmission direction. Because the velocity of propagating shear waves increases or decreases as a function of tissue elasticity and can be tracked in real-time, quantitative two-dimensional maps of shear wave velocity (SWV), or shear modulus, can be visualized (shear modulus is quadratically related to SWV) [14]. SWE is often used for internal medicine issues as a complementary diagnostic procedure. In addition, SWE serves as an evaluation procedure to determine the stage of disease and to track/ ensure the success of therapy [15]. The aim of the present study was to compare shear waves, muscle size, muscle length, and muscle diameter on the masseter muscle at rest and in function in patients with and without bruxism. As SWE is not currently used in the field or temporo-mandibular functional analyses, it might be helpful to visualize the altered masticatory muscles in terms of a minimally invasive procedure. Thus, the ultimate aim was to determine whether SWE is suitable for the diagnosis of bruxism.

## 2. Materials and Methods

Masseter muscles of both sides were investigated in 10 clinically confirmed bruxism patients with the conventional B-mode ultrasound and SWE. The bruxism patients were recruited from the TMD consultation hour at the Department of Prosthodontics at University Hospital Tübingen (mean age: 33.9 years ± 13.81 SD; 3 males, 7 female) who regularly attended a specific bruxism consultation in the local dental hospital and did not have a botulinum-toxin treatment within the past six months. For comparison, the masseter muscles of 10 healthy adult controls (HC; mean age: 23 years ± 2.28; 7 males, 3 female) were also measured. Neither bruxism patients nor healthy control had any implants or removable dentures. The HC group was only included if they could present a negative TMD-screening using the CMD-questionnaire from the German Society for Functional Diagnostics and Therapy (DGFDT) and had no other neurological, psychiatric, or psychosomatic disorders [16].

Each bruxism patient or HC was examined with B-mode Ultrasound (Canon Aplio i800 device; 14 MHz linear transducer, i14LX5/PLI-1205BX, Canon Medical Systems, Neuss, Germany) under the same conditions: sitting upright and in a relaxed position (Figure 1). First, the size of the muscle was measured in diameter (both relaxed and with maximal subjective bite force), and length and width on both sides to identify possible changes, such as hypertrophy in the muscle itself. This was followed by the actual SWV measurements, where we took three measurements each with three slightly different loci in the examined area of the muscle. The following SWE settings were used: size of region of interest (ROI): 2; ROI shape: radius, frame rate: 1; time smoothing: 0 (no time averaging); map type: speed (display of the shear wave velocity in meters per second).

In order to acquire a better overview of possible changes in masseter elasticity, SWE was performed under three conditions on all study subjects: relaxed musculus masseter (physiological rest position), with a subjective bite force of 50%, and in maximal mouth opening. For possible correlations, we also measured the incisor distance at maximal jaw opening. The positioning of the ultrasound was in the thickest area of the masseter muscle (Figure 1).

The statistical analysis of the collected data was performed with the Statistical Package for the Social Sciences (SPSS) version 27 for Windows, IBM Cooperation (Armonk, NY, USA). We began by checking for a normal distribution using the Shapiro–Wilk test. Here, normality was rejected due to significant results in SWE data for both the HC and bruxism patients. Therefore, non-parametrical testing with Mann–Whitney U test for the unpaired samples was applied to compare SWE levels of the bruxism patients with the HC in all three conditions: relaxed, 50% of maximal bite force, and maximal mouth opening. This was completed with the Wilcoxon for paired samples. These tests demonstrated whether there was a significant difference in SWE for the specific conditions with a significance level of *p* < 0.05. Moreover, the Spearman rho correlation coefficient was calculated using SWE and incisor distance at maximal jaw opening to investigate possible correlations between SWE and the limitations of jaw opening in bruxism patients. Box plots and graphs were created by JMP Statistical Package, Version 16.0.0 (SAS, Cary, NC, USA).

## 3. Results

The main results of the measurements are shown in Table 1. All healthy controls and patients met the inclusion criteria; bruxism patients were slightly older than healthy controls (bruxism patients—mean age: 33.9 years ± 13.81 SD, three males, seven females; healthy controls—mean age: 23 years ± 2.28, seven males, three females). Age and gender did not show any significant effect on the assessed variables in Table 1. Neither healthy controls, nor patients, reported any adverse effects during the study.

### 3.1. Measurement of the Musculus Masseter in the HC and Bruxism Patients

The measured sizes of the masseter muscle showed a significant difference in all planes: longitudinal (length), transverse (width), and sagittal (diameter). The median sizes of the HC were larger in diameter with relaxed jaw and maximal subjective bite and wider and longer than the sizes of the bruxism patients. Significance was proved in all dimensions by the Mann–Whitney U test (*p* < 0.01) (Figure 2 and Figure 3).

### 3.2. SWE in the HC and Bruxism Patients

For a relaxed and closed mouth, the SWE of bruxism patients was significantly higher than the SWE of the HC, i.e., median 1.91 m/s vs. 1.66 m/s, *p* < 0.01. When the patients were told to open their mouths as wide as possible, the HC group (3.53 m/s) had a significantly higher result in comparison with bruxism patients with a SWV of 2.89 m/s (*p* < 0.01). Our investigation with a subjective bite force of 50% showed no significant difference between the bruxism patients (2.77 m/s) and the HC (2.42 m/s; *p* < 0.722) (Figure 4).

### 3.3. Correlations, Maximal Incisor Distance

Potential correlations were further investigated using the measurements related to mouth opening. Here, incisor distance was measured after maximal opening of the mouth, which was significantly reduced in bruxism patients (mean 4.5 ± 1.2 cm) compared with the HC (mean 5.2 ± 0.4 cm) (*p* < 0.01). Using the Spearman rho coefficient, a correlation between the restricted mouth opening and the reduced elasticity of muscles in SWE results of Bruxism patients was proved (*p* < 0.01), whereas the correlation was not observed for the HC (*p* < 0.72) (Table 1).

## 4. Discussion

Currently, bruxism and consecutive muscular changes of the masticatory muscles, such as palpatory increased stiffness, muscle pain, and muscle hypertrophy or atrophy, are evaluated clinically, based on the subjective impression of the treating dentist, and there is a need for potential objective biomarkers for these common disorders. As SWE was already suggested to represent such an objective biomarker for muscle stiffness in the orofacial region [17,18,19,20], this study was conducted as a first step to evaluate whether SWE could be a potential new and objective diagnostic in bruxism.

SWE at rest was higher in bruxism patients compared with HC (median 1.91 m/s vs. 1.66 m/s, *p* < 0.01), which indicates that the stiffness of the masseter muscle was increased. Diagnostically, this has only been shown to be hardened by palpation of the masseter muscles. Thus, SWE could be an objective method to determine the stiffness of the masseter muscles. In addition, SWE could correlate with muscle tightness, as shown by the increased SWE in the bruxism group at a subjective bite force of 50%. There was a difference between bruxism patients (2.77 m/s) and the HC (2.42 m/s; *p* = 0.722), which suggests that the patients in the bruxism group have a more “trained” musculus masseter. However, it is noticeable that the masseter muscle is smaller in bruxism patients than in the HC. The bruxism patients probably bite harder than the HC as the SWE was higher, and SWE correlates positively with muscle force. This correlation was previously shown in the biceps brachii muscle [21]. Lower SWE values were shown by the bruxism patients (2.89 m/s) at max jaw opening compared with the HC (3.53 m/s; *p* < 0.01). This can be explained by the fact that the bruxism group had, on average, a lower incisor distance than the HC group. As a result, the musculus masseter was less tense, and the SWE was lower. The restricted jaw opening in bruxism patients could be caused by pain or by a change in the musculature. A possible cause would be a contraction of the muscle fibers. In the B-mode ultrasound examination, it could not be determined that the bruxism patients had larger muscles; they were even smaller in median length, width, and diameter than in the HC group in the patient clientele examined. Nevertheless, significant differences were found based on the SWE. Quantifying the size of a muscle is highly subjective and depends on the type of measurement. As a diagnostic measure for bruxism, the size measurement of the masseter muscle has limited applicability. An enlarged masseter muscle does not diagnostically mean that the patient has bruxism. As there is no direct correlation between muscle size and bruxism, muscle size should not be used as a diagnostic measure in bruxism patients. For example, the bony structure of the skull also plays a role in how the masticatory muscles are anatomically formed in size and shape [22,23]. The SWE seems to be more objective and provides information about the stiffness of the masticatory muscles. This diagnostic method can measure and determine the anatomical size in terms of the length, width, and diameter of the muscles. Our results demonstrated the good applicability of SWE in dental functional analyses for measuring the tension, contraction, and stiffness of the masseter muscle at rest and in function. This method can be used during diagnosis and in the course of therapy to assess the stiffness of the muscles at rest and in function. The means of therapy and the combination with physiotherapy can also be adjusted to this. Similarly, other changes in the muscle or other diseases can be excluded or confirmed diagnostically. The limiting factor of the study was the small number of patients, so, further investigations must be carried out. However, preliminary conclusions can be drawn that represent the results well. The next step will be a study with more representative results due to a higher number of patients and a follow-up during bruxism therapy. Therefore, this procedure can be used to examine not only the masticatory muscles but also the temporomandibular joint and adjacent structures, such as the disc and ligaments in the temporomandibular joint region [24,25,26].

External influences seem to play a role, such as in how patients deal with stress and everyday situations.

When evaluating the results, it must be taken into account that the functional movements (relaxed masticatory muscle, a subjective bite force of 50%, and active maximal mouth opening) correspond to the subjective sensation of the patient, and, thus, values of varying size or smallness can arise.

Nevertheless, our group results allow the identification of alterations in the bruxism group. The muscle measurements clarified the assumption that different muscle stresses have an effect on the muscle shape (width, length, and muscle diameter). There was evidence that the tension of the masseter muscle was greater in men and increases with age [20,27,28,29,30]. Moreover, when measuring the masseter muscle, it was found that the masseter muscle was smaller in bruxism patients and is subject to atrophy. This was not expected but may be related to the fact that the muscle fibers are shortened due to long-term contraction [31,32]. The musculus masseter in bruxism patients is shorter, not as wide, and has a smaller diameter than in healthy patients. Thus, an assumed hypothesis of muscle hypertrophy in bruxism in our patient cohort was not applicable. The cause could be a permanent contraction of the musculus masseter, which leads to a shortening of the muscle fibers. Similarly, the time of day of the survey may have had an effect on this result. The contraction of the musculus masseter could increase throughout the day due to stress, for example. Thus, there is greater muscle tension and muscle stiffness at rest. In muscle flexion at mouth opening, smaller SWV were measured in bruxism patients compared with the HC group, which was probably due to the shortened muscle fibers. Larger SWV at the musculus masseter occurred again at a subjective bite force of 50% in bruxism patients. The SWV at a subjective bite force of 50% was smaller in the HC. In addition, the bruxism patients had a lower incisor distance than the HC patients. This may be due to symptoms, such as muscle spasms, muscle pain, or temporomandibular joint problems. Because all these symptoms are subjective patient perceptions, a correlation to the resting values with relaxed musculus masseter could be conceivable. To analyze this in more detail would have required a patient survey at different time points and different degrees of bruxism.

There is a subjective and subconscious sensation in bruxism and craniomandibular dysfunction as well as a difference in muscle contraction during the day and at night, which has already been shown by several studies. At night, there is usually a stronger muscle contraction, which is related to teeth clenching and grinding [33,34,35]. Because the nocturnal contraction of the masticatory muscles and especially of the musculus masseter is greater at night, the nocturnal wearing of splints is recommended to alleviate symptoms, such as tooth wear and overloading of the temporomandibular joint. There are different approaches to alleviating symptoms. Occlusal splints are the most commonly used, as they decrease muscle activity [36]. Splints with an anterior jig are particularly recommended [37,38,39,40,41]. In addition, measurements, such as physiotherapy, manual therapy, physical therapy, stimulation current, or biofeedback, are recommended when prescribing a splint [42,43,44,45]. For severe muscle and joint pain, analgesics or botulinumtoxin are given. Analgesics have no effect on muscle activity, and botulinumtoxin can reduce the frequency of bruxism periods [46,47]. In some cases, psychotherapeutic measures are also indicated, which are intended to influence cognitive behavior [48].

For further studies, it would be interesting to measure SWV in bruxism patients under different measures of relief to examine the effectiveness of the different therapeutic approaches.

## 5. Conclusions

The study showed, preliminarily, that SWE diagnostic methods might be used as a potential tool for functional analysis in dentistry. The masticatory musculature, and especially the masseter muscle, could be measured at a relaxed position and in function, and tension could be detected by measuring the SWV. In addition, we partly detected the pathological changes in bruxism on the musculus masseter. SWE in bruxism is feasible and might be of potential use for diagnostics and monitoring, but we also highlighted important limitations and methodological considerations for future studies. The results shown need to be confirmed by a higher number of patients and SWE examination before and during bruxism therapy.

## Figures and Tables

**Figure 1 diagnostics-13-00276-f001:**
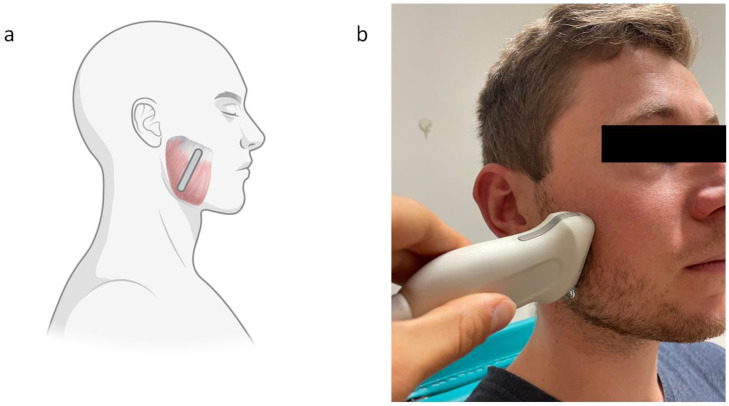
(**a**) Schematic illustration, and (**b**) clinical illustration, of the probe on the masseter muscle during a shear wave elastography examination.

**Figure 2 diagnostics-13-00276-f002:**
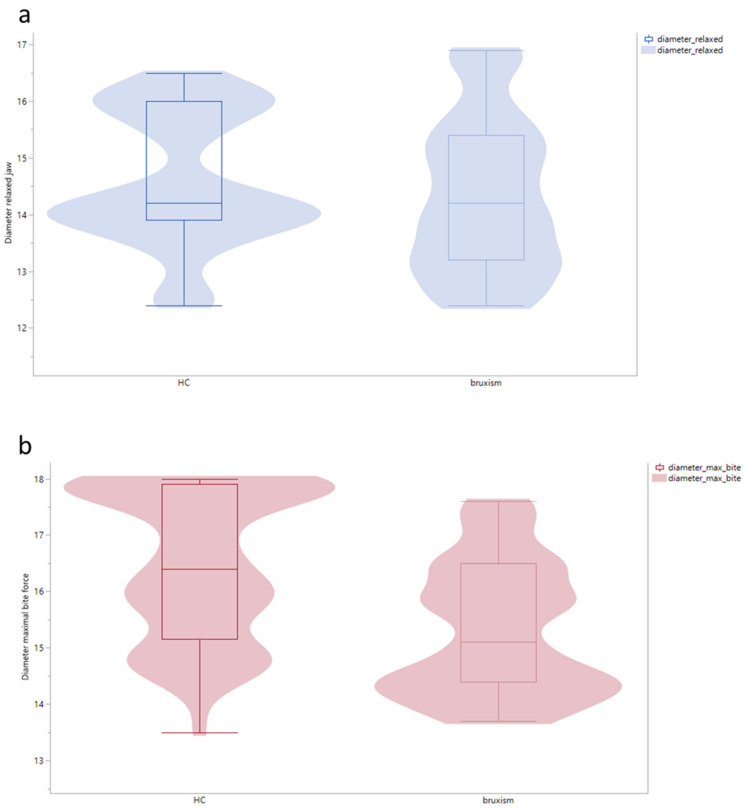
(**a**) Diameter relaxed, and (**b**) diameter with full force bite, of the musculus masseter in HC and bruxism patients (*p* < 0.01).

**Figure 3 diagnostics-13-00276-f003:**
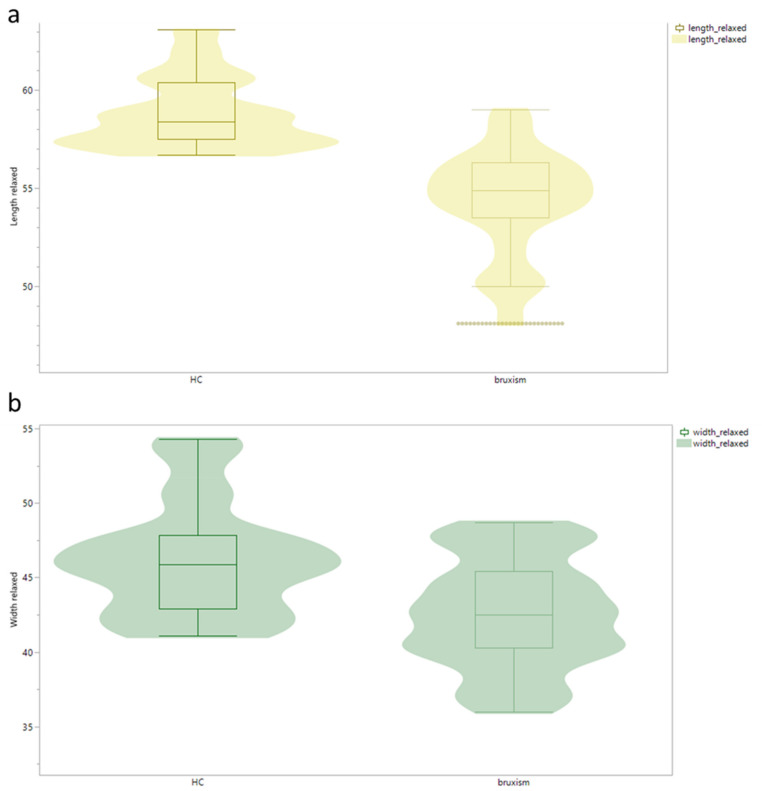
(**a**) Length, and (**b**) width, of the musculus masseter in HC and bruxism patients (*p* < 0.01).

**Figure 4 diagnostics-13-00276-f004:**
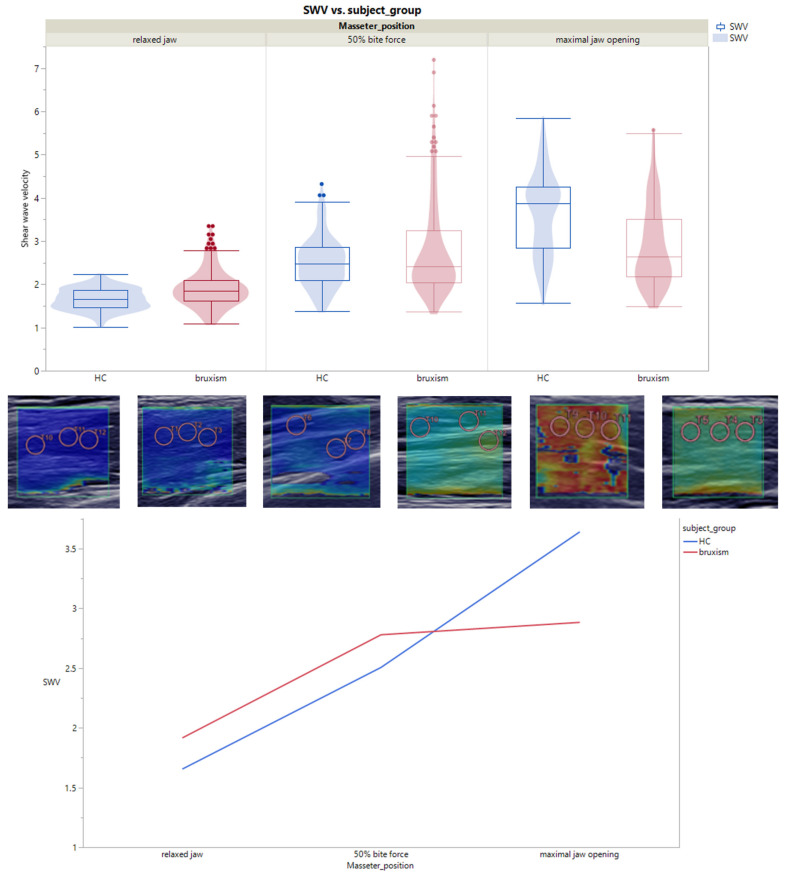
SWE of the musculus masseter by relaxed jaw (*p* < 0.01), 50% bite force (*p* = 0.722), and maximal jaw opening (*p* < 0.01) with the following settings: region of interest (ROI) size: 2; ROI shape: radius; frame rate: 1; time smoothing: 0 (no time averaging); map type: speed (display shear wave speed in meters per second).

**Table 1 diagnostics-13-00276-t001:** Results for the measurements of the musculus masseter, SWV, and maximal incisor distance of healthy controls and bruxism patients. Mean ± standard deviation, percentile 25%/50%/75% (range of all values).

Parameter	Healthy Control (HC)	Bruxism (B)	*p* B vs. HC
Diameter relaxed (mm)	14.6 ± 1.2, 13.9/14.2/16 (12.4–16.5)	14.3 ± 1.4, 13.2/14.2/15.4 (12.4–16.9)	<0.01
Diameter full force of contraction (mm)	16.5 ± 1.3, 15.2/16.4/17.9 (13.5–18.0)	15.3 ± 1.2, 14.4/15.1/16.5 (13.7–17.6)	<0.01
Width (mm)	46.3 ± 3.8, 42.9/45.9/47.9 (41.1–54.3)	42.6 ± 3.7, 40.3/42.5/45.4 (36.0–48.7)	<0.01
Length (mm)	58.9 ± 1.8, 57.5/58.4/60.4 (56.7–63.1)	54.3 ± 2.7, 53.5/54.9/56.3 (48.1–59.0)	<0.01
SWV relaxed (m/s)	1.66 ± 0.25, 1.5/1.66/1.87 (1.01–2.24)	1.91 ± 0.44, 1.62/1.86/2.1 (1.09–3.36)	<0.01
SWV 50% bite force (m/s)	2.42 ± 0.61, 2.09/2.48/2.87 (1.22–4.32)	2.77 ± 1.11, 2.01/2.39/3.22 (1.35–7.16)	0.722
SWV max opening (m/s)	3.53 ± 0.95, 2.85/3.89/4.25 (1.57–5.85)	2.89 ± 0.92, 2.17/2.64/3.5 (1.5–5.6)	<0.01
Maximal incisor distance (cm)	5.2 ± 0.4, 5/5.4/5.5 (4.5–5.8)	4.5 ± 1.2, 3.3/4.4/5.5 (2.3–5.8)	<0.01

## Data Availability

Raw data will be made available upon reasonable request.

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
