# Peer review of "Shear Wave Elastography in Bruxism—Not Yet Ready for Clinical Routine"

_diagnostics, 2023, doi:10.3390/diagnostics13020276_

Round 1

Reviewer 1 Report

This study represents a novel shear wave application in biomedical field, however, I have several major issues to be taken into account before to proceed with the review process:

1. The introduction needs more references, from the point of view of biomechanics and physiological background of the disease and why is quite significant evaluate whit sw elastography this issue.

2. It seems that sample size have been chosen without a specific statistical criteria it seems to be an exploration due to the availability of the patients. I recommend to remark that it is a preliminar conclusion. Maybe a paired test with for example a follow up is interesting to consider more results now or in the future. 

3. why rho correlation is considered? the main conclusion is to infer the main differences between several sw measurements and healthy or bruxism groups.

4. In my opinion, figure 2 confidence intervals are recommended.

5. A paragraph mentioning the details of the ultrasonic setup according with previous setups considered by these authors or others and the comparison with the usual techniques or similars is needed.

6. How this study would improve the diagnosis or therapeutic implications of this disease? A brief conclusion about this would be interesting in the discussion. 

7. What is the perspective or next step?

Author Response

This study represents a novel shear wave application in biomedical field, however, I have several major issues to be taken into account before to proceed with the review process:

  1. The introduction needs more references, from the point of view of biomechanics and physiological background of the disease and why is quite significant evaluate whit sw elastography this issue.
  • Answer: We thank the reviewer for pointing this importance out! We added/modified the introduction extensively: “20% of adults have bruxism symptoms, which are expressed differently in the symptomatic [3]. The rhythmic masticatory muscle activity is particularly high in bruxism at night, for this reason it is often referred to as sleep bruxism, which is said to be caused by the transient activity of the brainstem arousal-reticular ascending system [4, 5]. It is a disease of the oral, maxillofacial area, which has various effects and symptoms: Besides pain during jewing, tooth attrition, muscle changes such as hypertrophy and the main symptom is the stiffness of the masticatory muscles resulting in a limited mouth opening, mostly [6-8].“ and “Ultrasound can easily visualize the size change of the masseter muscles for diagnosis and therapy [12].

  1. It seems that sample size have been chosen without a specific statistical criteria it seems to be an exploration due to the availability of the patients. I recommend to remark that it is a preliminar conclusion. Maybe a paired test with for example a follow up is interesting to consider more results now or in the future. 
  • Answer: We agree and therefore write this now more clearly The limiting factor of the study is the small number of patients, so that further investigations must be carried out. However, preliminary conclusions can be drawn that represent the results well. The next step is a study with more representative results due to a higher number of patients and a follow-up during in bruxism therapy.“

  1. why rho correlation is considered? the main conclusion is to infer the main differences between several sw measurements and healthy or bruxism groups.
  • Answer: We thank the reviewer for the watchful eye and believe we put the reason in the introduction „Moreover, the Spearman-Rho correlation coefficient was calculated using SWE and incisor distance at maximal jaw opening to investigate possible correlations between SWE and the limitations of jaw opening in bruxism patients

  1. In my opinion, figure 2 confidence intervals are recommended.
  • Answer: Altough we value the suggestions of the reviewer, we believe that the common boxplot with median, IQR and Min/Max are sufficient.

  1. A paragraph mentioning the details of the ultrasonic setup according with previous setups considered by these authors or others and the comparison with the usual techniques or similars is needed.
  • Answer: We thank the reviewer for this important note and agree with him: Therefore the actual SWE settings are written “The following SWE settings were used: size of region of interest (ROI): 2; ROI shape: radius, frame rate: 1; time smoothing: 0 (no time averaging); map type: speed (display of the shear wave velocity in meters per second).“ as well as the setup “Each bruxism patient or HC was examined with B-mode Ultrasound ( Canon Aplio i800 device; 14 MHz linear transducer, i14LX5/PLI-1205BX, Canon Medical Systems, Neuss, Germany) under the same conditions: sitting upright and in a relaxed position (Fig. 1). First, the size of the muscle was measured in diameter—both relaxed and with maximal subjective bite force—length, and width on both sides to identify possible changes, such as hypertrophy in the muscle itself. This was followed by the actual SWV measurements, where we took three measurements with each three slightly different loci in the examined area of the muscle.“

  1. How this study would improve the diagnosis or therapeutic implications of this disease? A brief conclusion about this would be interesting in the discussion. 
  • Answer: We agree and thank the reviewer for this suggestion This method can be used during diagnosis and in the course of therapy to assess the stiffness of the muscles at rest and in function. The means of therapy and the combination with physiotherapy can also be adjusted to this. Likewise, other changes in the muscle or other diseases can be excluded or confirmed diagnostically. The limiting factor of the study is the small number of patients, so that further investigations must be carried out. However, preliminary conclusions can be drawn that represent the results well. The next step is a study with more representative results due to a higher number of patients and a follow-up during in bruxism therapy.“

  1. What is the perspective or next step?
  • Answer: Added to the previous suggestion.

Reviewer 2 Report

Congratulations to the authors of this article. It is a very interesting article from a clinical and research point of view, and I am sure that new research on this subject will emerge from this study. I have some doubts about the paper, which I will explain below:

-          Figures should be numbered as they come out of the text.

-          Table titles are usually placed above the table

-          You could put the data on both sides in both healthy subjects and subjects with bruxism. In this way, you could have compared the change between left and right masseter.

-          In the results section, it would give written information such as effect sizes, correlations, if there have been any adverse effects, if all the patients have met the inclusion and exclusion criteria, mean differences...

-          The discussion section should be written in more detail. It is written superficially. Possible mechanisms can be explained.

Author Response

Congratulations to the authors of this article. It is a very interesting article from a clinical and research point of view, and I am sure that new research on this subject will emerge from this study. I have some doubts about the paper, which I will explain below:

  • Figures should be numbered as they come out of the text.  
  • Answer: We thank the reviewer for this hint and have changed it accordingly.
  • Table titles are usually placed above the table
  • Answer: Has been revised and modified.
  • You could put the data on both sides in both healthy subjects and subjects with bruxism. In this way, you could have compared the change between left and right masseter.
  • Answer: Here is a possible table with average values for both sides (left and right). There is no decisive difference, so we have refrained from showing the values in the article.

Parameter

 Healthy Control left (HCR)

 Healthy Control right (HCL)

Bruxism left

(BL)

  Bruxism right

           (BR)   

Diameter relaxed (mm)

14.6

14.1

14.1

14.6

Diameter full force of contracion (mm)

           16.1

           16.8

14.9

15.9

Width (mm)

46.9

46.2

42.6

42.5

Length (mm)

59.1

58.8

54.4

54.3

SWV relaxed (m/s)

1.68

1.69

1.94

1.89

SWV 50% bite force (m/s)

2.47

2.41

2.72

2.79

SWV max opening (m/s)

3.8

3.3

2.86

3.12

Maximal incisor distance (cm)

5.2

4.5

  • In the results section, it would give written information such as effect sizes, correlations, if there have been any adverse effects, if all the patients have met the inclusion and exclusion criteria, mean differences...
  • Answer: We thank the reviewer for this advise and added “All healthy controls and patients met the inclusion criteria; bruxism patients were slightly older than healthy controls (bruxism patients: mean age: 33.9 years ± 13.81 SD; 3 males, 7 female – healthy controls: mean age: 23 years ± 2.28; 7 males, 3 female). Age and gender did not show any significant effect on the assessed variables in Table 1. Neither healthy controls, nor patients reported any adverse effects during the study.”

  • The discussion section should be written in more detail. It is written superficially. Possible mechanisms can be explained.
  • Answer: We thank the reviewer for this suggestion; we rewrote the large parts of discussion section.

Round 2

Reviewer 1 Report

All questions and suggestions have been adequately answered